# Six Decades of Dopamine Hypothesis: Is Aryl Hydrocarbon Receptor the New D2?

**DOI:** 10.3390/reports6030036

**Published:** 2023-08-01

**Authors:** Adonis Sfera

**Affiliations:** 1Paton State Hospital, 3102 Highland Ave, Patton, CA 92369, USA; adonis.sfera@dsh.ca.gov; 2School of Behavioral Health, Loma Linda University, 11139 Anderson St., Loma Linda, CA 92350, USA; 3Department of psychiatry, University of California, Riverside 900 University Ave, Riverside, CA 92521, USA

**Keywords:** dopamine hypothesis, microbial translocation, antipsychotic drugs

## Abstract

In 1957, Arvid Carlsson discovered that dopamine, at the time believed to be nothing more than a norepinephrine precursor, was a brain neurotransmitter in and of itself. By 1963, postsynaptic dopamine blockade had become the cornerstone of psychiatric treatment as it appeared to have deciphered the “chlorpromazine enigma”, a 1950s term, denoting the action mechanism of antipsychotic drugs. The same year, Carlsson and Lindqvist launched the dopamine hypothesis of schizophrenia, ushering in the era of psychopharmacology. At present, six decades later, although watered down by three consecutive revisions, the dopamine model remains in vogue. The latest emendation of this paradigm proposes that “environmental and genetic factors” converge on the dopaminergic pathways, upregulating postsynaptic transmission. Aryl hydrocarbon receptors, expressed by the gut and blood–brain barrier, respond to a variety of endogenous and exogenous ligands, including dopamine, probably participating in interoceptive awareness, a feed-back loop, conveying intestinal barrier status to the insular cortex. The conceptualization of aryl hydrocarbon receptor as a bridge, connecting vagal terminals with the microbiome, may elucidate the aspects of schizophrenia seemingly incongruous with the dopamine hypothesis, such as increased prevalence in urban areas, distance from the equator, autoantibodies, or comorbidity with inflammatory bowel disease and human immunodeficiency 1 virus. In this review article, after a short discussion of schizophrenia outcome studies and insight, we take a closer look at the action mechanism of antipsychotic drugs, attempting to answer the question: do these agents exert their beneficial effects via both dopaminergic and nondopaminergic mechanisms? Finally, we discuss potential new therapies, including transcutaneous vagal stimulation, aryl hydrocarbon receptor ligands, and restoring the homeostasis of the gut barrier.

## 1. Introduction

The dopamine hypothesis (DH) of schizophrenia (SCZ), launched by Carlsson and Lindqvist in 1963, surmises that the postsynaptic blockade of dopamine (DA) receptors is responsible for the beneficial effect of antipsychotic drugs in SCZ and SCZ-like disorders [1]. Arvid Carlson’s experiments with DA depletion by reserpine, followed by chlorpromazine inhibition of dopaminergic transmission, led to a better understanding of the pathology at work in Parkinson’s disease (PD) and acute psychosis [2]. Carlsson and Lindqvist surmised that excessive DA activation of the postsynaptic DA type 2 receptors (D2R) in the central nervous system (CNS) triggered psychotic symptoms, while dopaminergic blockade of these receptors comprised the remedy [3]. This model was further substantiated by the 1970s observation that amphetamine enhanced dopaminergic signaling, often exacerbating psychotic symptoms [4]. However, the realization that antipsychotic drugs exert properties, seemingly unrelated to DA, such as antimicrobial, antiviral, antiproliferative, cell cycle arrest, autophagy activation, alteration of iron metabolism, DNA methylation, and telomere elongation, led to the third DH revision, which emphasized genetic and epigenetic input in dopaminergic transmission [5,6,7,8,9,10].

Aside from the antipsychotic drugs, several characteristics of SCZ itself may be incongruous with the DH, including increased prevalence in industrialized countries and urban areas, variance with latitude, autoantibodies, and high comorbidity with inflammatory bowel disease (IBD) and human immunodeficiency virus 1 (HIV-1) (Table 1). Moreover, excessive DA in the CNS would likely promote euphoria, motivation, and heightened alertness instead of hallucinations or delusions. Furthermore, DH equates acute psychosis with SCZ, an assumption not always shared by clinicians, many of whom conceptualize positive symptoms of SCZ as epiphenomena of this pathology [11].

Aryl hydrocarbon receptor (AhR), a cytosolic transcription factor, initially described as the dioxin receptor, responds to numerous exogenous and endogenous ligands, inducing both immune tolerance of gut microbes as well as their prompt elimination upon translocation into host tissues [43,44]. In the cytosol, AhR is bound by two heat shock protein 90 (HSP90) chaperones, molecules recently identified as both SCZ and Parkinson’s disease (PD) targets [45,46,47,48,49,50,51,52]. Other AhR ligands significant for SCZ include DA, carbidopa, clozapine, melatonin, serotonin, microbial phenazines, and various pollutants, linking endogenous and exogenous molecules to this pathology [53,54,55,56].

At the subcellular level, AhR, a signal transducer and activator of transcription 3 (STAT3) and interleukin 22 (IL-22), communicates with both the central cholinergic system and the microbiome, relaying the “gut status” to the insular cortex (IC) [57,58,59]. Impaired AhR signaling was associated with IBD and HIV, disorders highly comorbid with SCZ, marked by a dysfunctional gut barrier, suggesting that microbial translocation may be the common denominator of these pathologies [60,61]. For example, upregulated plasma levels of immunoglobulins M and/or A against *Hafnei alvei, Pseudomonas aeruginosa, Morganella morganii, Pseudomonas putida,* and *Klebsiella pneumoniae* were found in SCZ patients with negative symptoms, further linking this disorder to microbial translocation [12]. Furthermore, poor insight or anosognosia, a characteristic SCZ feature, present in up to 98% of patients, was associated with intestinal-inflammation-induced IC pathology [62,63,64,65].

We hypothesize that gut barrier disruption enables the migration of microbes or their components into the host systemic circulation, eventually reaching the brain, generating neuroinflammation and psychotic symptoms. This model is based on the following findings:In SCZ, IBD, and HIV-1, the blood microbiome exhibits an increased level and diversity of intestinal microbes, suggesting translocation from the gut [66,67,68].Monocytosis and elevated microbial translocation markers, including lipopolysaccharide (LPS), LPS-binding protein (LBP), and soluble CD14 (sCD14), were documented in SCZ, HIV, and IBD, indicating translocation [69,70,71,72,73,74].Elevated cell-free DNA (cfDNA) in SCZ, HIV, and IBD likely reflects gut barrier and BBB disruption [75,76,77].Various “autoantibodies” detected in SCZ, IBD, and HIV are likely conventional immunoglobulins elicited by the translocated microbes and/or their components [78,79,80,81].

In this review article, after a brief discussion of SCZ outcome studies and anosognosia, we discuss the action mechanism of antipsychotic drugs, interrogating the nondopaminergic actions of these agents. Finally, we discuss new therapies, including transcutaneous vagal stimulation, aryl hydrocarbon receptor ligands, and recombinant IL22-mediated gut barrier rehabilitation. Less emphasis will be placed on the SCZ topics discussed in other articles, such as premature cellular senescence, ferroptosis, DNA methylation, innate lymphoid cells (ILCs), and lactylation [82,83].

## 2. DH and SCZ Outcome Studies

SCZ is a complex, multifactorial, and partially understood disorder marked by episodic symptomatic relief intertwined with periods of exacerbation and relapse [84]. The outcome of this illness is variable and difficult to measure as the conceptualization of functional recovery may differ among people [84]. For example, phrases like improved social and cognitive functioning, greater autonomy to manage one’s own life, remission of positive and negative symptoms, better quality of life, or reintegration into the community may raise different expectations in each involved party, including patients, families, clinicians, or consumer advocates, indicating that recovery in SCZ is a complex and often subjective process [85,86,87,88,89,90]. In contrast to recovery, remission is a more concrete outcome measure that requires six months of minimal symptomatology without a full return to the premorbid level of functioning [91]. Despite the fact that remission is more attainable than recovery, most clinicians and family members would agree that living independently, maintaining stable employment or going to school, living an optimistic and hopeful life, and dating or getting married are goals rarely accomplished by patients with chronic SCZ even when adherent to treatment [92,93,94]. For example, novel studies have shown that 33% of SCZ patients relapse during the first 12 months after an initial psychotic episode, 26% remain homeless at 2 years of follow-up, while 5 years after the first psychotic outbreak, only 10% are employed [95,96,97]. Recovery at 15 and 25 years of follow-up is marginally better, at 16%, indicating that sustained recovery in chronic SCZ is infrequent [98]. Indeed, only 13.5% of patients meet recovery criteria at any point in time after the first psychotic episode [99].

Surprisingly, longitudinal studies looking at the overall SCZ recovery rate during the 20th century found little change since 1900, and, according to some studies, a steadily deteriorating outcome [99,100,101]. Moreover, contrary to the expectations of most researchers and clinicians, the proportion of patients with “good” outcomes has not increased in the decades following the discovery of antipsychotic drugs [7,90,102,103]. For example, a large meta-analysis of 114 follow-up studies by Warner R. looked at the time period from 1900 to 1996 and found the following rates of complete recovery and employment [101] (pp. 74):1901–1920, 20% complete recovery, 4.7% employed;1921–1940, 12% complete recovery, 11.9% employed;1941–1955, 23% complete recovery, 4.1% employed;1956–1975, 20% complete recovery, 5.1% employed;1976–1995, 20% complete recovery, 6.9% employed.

These data led the author to conclude that “recovery rates for patients admitted following the introduction of the antipsychotic drugs are no better than for those admitted after the Second World War or during the first two decades of the twentieth century” (page 78). Moreover, the employment rate of patients with SCZ has been decreasing steadily over the past 50 years in industrialized countries, a finding congruent with Warner’s recovery data [104,105]. Furthermore, unlike public hospitals for tuberculosis or leprosy, discontinued for over half a century, long-term state institutions for the treatment of SCZ and like disorders continue to exist, representing a proof of concept that sustained recovery from these illnesses remains unsatisfactory [90,100].

Neuroimaging SCZ studies correlate well with the recovery and employment data, showing progressive, treatment-independent cortical gray matter loss, suggesting that the evolution of this disorder toward disability, cognitive deficit, and early neurodegeneration may be undeterred by the available therapies [86,106,107,108,109,110]. However, several neuroimaging studies in SCZ patients treated with second-generation antipsychotic drugs found delayed rates of gray matter loss, indicating that more outcome studies are needed in patients on atypical antipsychotics [111,112].

Taken together, although there is no doubt that antipsychotic drugs are extremely effective for the treatment of acute psychosis, chronic SCZ is less amenable to recovery, a finding in line with Kraepelin’s initial observations of steady progression to premature dementia [113]. In summary, the reasons DH may require further revisions include the following:Continued need for long-term public institutions for the treatment of chronic mental illness.DA blockers may not alter the progression of SCZ toward disability and cognitive deficit.Several SCZ characteristics are difficult to reconcile with the DH.Life-long gray matter loss occurs despite treatment with DA blockers.Upregulated DA in the CNS would be expected to result in euphoria, increased motivation, and alertness rather than hallucinations or delusions.Anosognosia, the most common symptom of SCZ, is infrequently influenced by DA-blocking treatments.

## 3. Insight vs. Anosognosia, Lessons from COVID-19 and HIV

Poor insight or anosognosia is a cardinal symptom of SCZ, occurring in up to 98% of patients, which may not respond adequately to antipsychotic drugs [19]. Insight has been associated with interoceptive awareness and perception of internal body cues, such as heartbeat, respiration, intestinal function, position of limbs, and “self” vs. “non-self”, a process impaired in many neuropsychiatric conditions, including traumatic brain injury (TBI), stroke, and SCZ [20,21]. Novel studies have associated IC, an area previously implicated in SCZ, with both anosognosia and error awareness [22,23,24,25,63].

Communication between IC and vagus nerve (VN) was described more than three decades ago, and recent studies have not only confirmed the existence of this dialog but also found that transcutaneous auricular VN stimulation (taVNS) can enhance interoceptive awareness, suggesting potential application in SCZs [114,115]. Others have shown that gut microbiota can alter IC connectivity, indicating that this area specializes in processing GI tract interoceptive input [116]. Interestingly, a study found that risperidone can upregulate IC connectivity, suggesting that second-generation antipsychotic drugs may influence insight more than the first-generation compounds [117]. In this regard, recent studies have reported that GI tract inflammation activates IC, while electrical stimulation of IC exerts an anti-inflammatory GI tract response, suggesting a potential therapeutic strategy for IBD [62,118]. Along this line, a preclinical study found that memories of previous intestinal inflammations (induced in rodents by mixing dextran sodium sulfate (DSS) with drinking water) are stored in IC, and stimulation of this area, weeks or months later, can trigger intestinal inflammation in the absence of DSS [62,118]. This study is in line with other research data on interoceptive awareness and IC, likely pinpointing the neuroanatomical location of an “insight center” [64,118,119]. Indeed, functional neuroimaging and connectivity studies have demonstrated that the awareness of error, insight, and the position of limbs is regulated by the IC neuronal networks [120]. Moreover, neuroimaging studies in patients with Chron’s disease are in line with preclinical data, linking IC connectivity to IBD [62,120,121]. Interestingly, activation of IC was shown to increase the abundance of gut *Prevotella* and *Bacteroides species*, indicating that insular neurons are capable of regulating microbiota composition [116]. 

### 3.1. Mononuclear Cells and Insight

Novel data have shown that AhR, the master regulator of peripheral mononuclear cells, drives the differentiation of monocytes into dendritic cells vs. macrophages/microglia, therefore altering the peripheral blood level of these cell types [122]. 

Studies during the COVID-19 pandemic found that the SARS-CoV-2 virus can induce cognitive deficit of which the patients themselves were unaware, known as anosognosia for impaired memory. Moreover, anosognosia in these individuals was associated with peripheral monocytosis (defined as 7.35% or more of the total number of leukocytes), suggesting that infections can alter insight, while monocyte count could represent a biomarker for interoceptive awareness [123]. The correlation between monocytosis and anosognosia is not new or limited to COVID-19, as it was reported earlier in the context of HIV-associated neurocognitive disorder (HAND), a condition marked by impaired insight [124]. This is significant as monocyte levels are elevated in SCZ, highlighting a potential general marker of anosognosia in neuropathology [125,126,127]. For example, a recent study found that patients in the preclinical phase of SCZ exhibited monocytosis and increased microbial translocation markers, further linking gut microbes with this pathology [128]. Moreover, upregulated blood monocytes were documented during the transition phase from mild cognitive impairment (MCI) to Alzheimer’s disease (AD), connecting bacterial translocation with neurodegenerative disorders [129,130]. Indeed, microbes and LPS were found in AD brains, further linking bacteria to cognition and insight [131,132,133]. Along this line, peripheral monocytes were shown to infiltrate the CNS and differentiate into microglia, cells that under pathological circumstances can engage in the aberrant phagocytosis of healthy neurons, predisposing to neurodegeneration [134,135]. For example, SCZ-related cognitive deficit, probably including anosognosia, was associated with the engulfment of intact synapses and dendritic spines by abnormally activated microglia [136,137].

### 3.2. Cholinergic Anti-Inflammatory Pathway and Insight

Discovered in 2000 by Borovikova et al. [138], the cholinergic anti-inflammatory pathway (CAP) is a brain-to-periphery neural loop mediated by alpha7 nicotinic acetylcholine receptors (α7nAChRs) expressed by neurons, immune cells, and intestinal epithelial cells (IECs) (Figure 1). CAP likely participates in interoceptive awareness and regulates intestinal permeability, reporting on the gut status, via VN, to the IC [138,139]. For example, addiction medicine studies have shown that both acetylcholine (ACh) and nicotine can activate IC, ameliorating anosognosia, further linking insight to CAP [140,141,142]. Moreover, CAP likely mediates “cholinergic behavior” (addiction, emotion, and motivation) after activation by ACh or nicotine, probably linking tobacco use by SCZ patients to improved awareness and insight [143,144,145,146,147,148,149]. Indeed, nicotine was demonstrated to improve spatial neglect in stroke patients, suggesting that anosognosia is driven, at least in part, by the cholinergic pathways [150,151] (Figure 1). 

Several novel studies have found that nicotine (and probably ACh) stimulates AhR via STAT3, engendering a “molecular interoceptive system” which connects to IC via VN [151]. Within this system, AhR receives input from the microbiome as well as the environment (via xenobiotics, toxins, and photo-metabolites) and relays these data to the CNS for processing and discernment, completing the interoceptive loop [152,153]. 

At the subcellular level, phosphorylated STAT3 (pSTAT3) bridges the gap between ACh, AhR, and IL-22, which closes the interoceptive circuit by generating more pSTAT3 and repeating the cycle [17,122,154,155]. Indeed, the AhR/STAT3/IL-22 axis upregulates *Bifidobacterium* and *Lactobacillus* spp., microbes known for producing IL-10, upregulating pSTAT3 further (Figure 1).

The barrier-protective role of IL-22 surfaced during the 1980s HIV-1 epidemic, an infection known for disrupting the intestinal barrier, promoting microbial translocation [156,157]. IL-22 is regulated by AhR/STAT3 and is produced by several types of lymphocytes, including T helper (Th) 17 cells, γδ T cells, natural killer cells (NKCs), and innate lymphoid cells (ILC) [158]. IL-22 plays a key role in preventing premature cellular senescence and thymic involution, a pathology documented in SCZ [159,160]. 

Taken together, interoceptive awareness is driven by the IC via CAP. At the subcellular level, the “molecular insight” is mediated by the AhR/STAT3/IL-22 axis, connecting the microbiome to the IC.

## 4. DH-Incongruent SCZ Features

Antipsychotic drugs are extremely efficacious for the treatment of acute psychosis and will likely remain the gold standard of psychiatric practice worldwide for the foreseeable future. However, in patients with chronic SCZ, the response to these drugs is less substantial as roughly one out of every seven patients recover, indicating a likely etiopathogenetic difference between acute psychosis and chronic schizophrenia [102,161,162,163]. Moreover, unlike the discovery of antibiotics which rendered inpatient public institutions for communicable diseases obsolete, the advance of antipsychotic drugs produced a less dramatic effect and only at the expense of high rates of incarceration and homelessness [90,164,165,166].

The gut microbiome is separated from the rest of the body by a single layer of IECs that microbes cross routinely during development to “educate” the host immune system in tolerating commensal flora. However, translocation during adult life, depending on the species, triggers inflammation and immunogenicity [167].

### 4.1. Markers of Gut Barrier Dysfunction

SCZ has been associated with microbial translocation, likely promoted by premature senescence and the accumulation of senescent, damaged cells at the gut barrier and BBB [13,168,169,170]. In SCZ, the pilling up of senescent cells is likely induced by accelerated tissue aging and/or impaired autophagy [171,172]. Aging cells are marked by short telomeres, replication arrest, active metabolism, and a detrimental secretome, known as senescence-associated secretory phenotype (SASP) that can spread senescence throughout the body [173]. In addition, senescent and defective cells can easily disintegrate, releasing immunogenic intracellular molecules, which can trigger autoimmune inflammation and barrier disruption [174]. For example, cell-free DNA (cfDNA), an emerging SCZ marker, was documented in IBD, an SCZ comorbid autoimmune disorder [175,176]. Interestingly, gut microbiota releases microbial cell-free DNA (cfmDNA) that can be detected in the peripheral blood, indicating a potential measurable marker of intestinal barrier integrity [177]. For example, elevated *Bacteroidetes* and *Firmicutes* cfmDNA, documented in IBD, may also be a marker of SCZ with a potential role in screening individuals at high risk [178,179]. To our knowledge, there are no studies of cfmDNA in SCZ; however, elevated cfDNA was demonstrated in patients with a first episode of SCZ, linking this pathology once more to gut barrier disruption [176,180].

### 4.2. Autoantibodies as Markers of Microbial Translocation

Over the past two decades, numerous “autoantibodies”, including anti-N-methyl-D-aspartate (NMDA) receptor antibodies, have been documented in SCZ, comprising phenomena difficult to reconcile with the DH. For example, the translocation of *Corynebacterium glutamicum,* a major glutamate producer, may account for NMDA receptor “autoantibodies”, documented in SCZ [181,182] (Table 2). However, instead of reflecting autoreactivity, these “autoantibodies” may be conventional immunoglobulins directed at translocated microbial antigens. 

Moreover, rather than “autoantibodies”, immunoglobulins against *Escherichia coli* (*E. coli*) proteins yjjU, livG and ftsE, found in many SCZ patients, may reflect *E. coli* translocation through the dysfunctional intestinal barrier associated with this disorder [40]. Furthermore, γ-Aminobutyric acid (GABA)-generating *Pseudomonas fluorescens* may, upon migration into host systemic circulation, elicit anti-GABA antibodies, immunoglobulins documented in patients with SCZ [41,42]. As this topic was discussed elsewhere, it will not be explored in more detail here [183].

### 4.3. AhR, and Antipsychotic Drugs

Discovered in 1976, AhR is a ligand-activated transcription factor and dioxin receptor, which regulates numerous cellular processes in response to exogenous and endogenous signals [184,185,186]. Currently, AhR is believed to be activated in a “tissue-specific” and “context-specific” manner which is not entirely clear, and more studies are needed to elucidate the ligands at this important receptor [187]. 

Under physiological circumstances, HSP90 retains AhR in the cytoplasm, preventing its ingress into the nucleus. HSP90 disassociation from AhR reveals a specific motif (adjacent to valine 647) that when exposed, promotes AhR nuclear entry, followed by either silencing or expression of several genes, possibly including the SCZ risk genes [188,189,190]. Indeed, the AhR/HSP90 complex has been implicated in psychosis and may exhibit opposite effects when residing in the cytoplasm vs. the nucleus. In general, high-affinity AhR ligands, primarily dioxin and plasticizers, induce neuronal apoptosis and cognitive deficit, while low-affinity AhR ligands and partial agonists or antagonists, such as quercetin, apigenin, or campherol, elicit neuroprotective effects [26,191,192]. Interestingly, aripiprazole-activated HSP90 has been associated with neurite outgrowth, suggesting involvement in neuroplasticity and cognition [193,194] (Figure 2). Moreover, both DA and clozapine are AhR ligands, while HSP90 has been involved in PD by inducing damage in midbrain DA neurons, suggesting that this complex plays a major role in neuropathology, probably accounting for the higher prevalence of SCZ in urban areas [53,195]. On the other hand, HSP90 inhibitors have emerged as therapies for neurodegenerative diseases and cancer [196]. Since DA is an AhR agonist, DA inhibition by antipsychotic drugs may have an antagonistic effect on the downstream AhR. This effect may strengthen the stability of AhR/HSP90 binding, maintaining this complex in the cytoplasm, thus preventing the transcription of SCZ risk genes [197].

In the gut, AhR is activated by tryptophan-derived microbial metabolites and participates in numerous physiological and pathological processes. Dietary AhR ligands promote STAT3 phosphorylation (pSTAT3), inducing the IL-22, the “guardian” of the gut barrier [198]. AhR ligands, including DA, serotonin, and melatonin, may play a major role in interoceptive awareness [63,64,65].

### 4.4. AhR and Environmental Pollutants

Exposure to various environmental pollutants during development has been associated with SCZ, likely accounting for the higher prevalence of this disorder in individuals born or raised in urban areas [27,28]. Environmental pollutants, including dioxin, bind with high affinity to AhR, inducing neuronal apoptosis and cognitive dysfunction. For example, prenatal exposure to diethylhexyl phthalate (DEHP), a widely used plasticizer, has been shown to disrupt the thymus, inducing premature involution of this gland, a pathology encountered in SCZ [29,30,199,200]. In addition, this pollutant triggers gut barrier disruption and was linked to autism spectrum disorder (ASD), suggesting likely involvement in SCZ [31,32]. Moreover, DEHP has been implicated in ferroptosis, a nonapoptotic cell death, triggered by lipid peroxidation in the context of elevated intracellular iron levels and impaired redox systems [201]. Indeed, lipid peroxidation is accelerated by pollutants, including the atmospheric fine particulate matter (FPM), previously linked to neuropathology, including SCZ [33,34]. Ferroptosis and lipid peroxidation in biological barriers promote translocation-related pathology, including SCZ.

### 4.5. AhR and Latitude Variance

The increasing prevalence rate of SCZ with distance from the equator indicates that daylight duration and light-activated AhR ligands play a major role in the pathogenesis of this disorder. The most studied sunlight-dependent AhR ligands include vitamin D3, and tryptophan photo-metabolites, such as 6-formylindolo [3,2-b]carbazole (FICZ), play a key role in latitude-dependent SCZ prevalence [35,202,203]. Upon vitamin D3/AhR binding, the receptor/ligand complex is shuttled into the nucleus where the transcription of multiple genes is initiated or suppressed [36,37,38]. In addition, FICZ upregulates IL22, protecting the gut barrier and BBB, likely explaining the lower prevalence of SCZ in warm climates [35,37]. Moreover, since latitude drives the diversity of gut microbiota, industrialized countries, found primarily at higher latitudes, exhibit less diverse microbes compared to equatorial areas of the globe [39].

Taken together, gut barrier disruption and microbial translocation, a common feature of SCZ, likely explain the association of this illness with being born and raised in urban environments as well as latitude variance, placing AhR at the very center of this pathology.

## 5. Non-Dopaminergic Antipsychotic Mechanisms of Neuroleptic Drugs

Several properties of antipsychotic agents appear difficult to reconcile with the DH, even when taking into consideration environmental and genetic factors [109]. For example, the antimicrobial properties of these drugs could alleviate psychosis by eliminating translocated microbes and subsequent inflammation [204]. The non-dopaminergic, therapeutic properties of antipsychotic drugs are likely driven by AhR, a protein involved in the regulation of antimicrobial, antiviral, and antiproliferative host defenses as well as pathogen clearance via reactive oxygen species (ROS), or autophagy [205,206,207]. For example, AhR-induced ROS can ameliorate psychotic symptoms by clearing intracellular pathogens, including *Toxoplasma gondii*, a SCZ-associated parasite [208]. In addition, neuroleptic drug-activated autophagy and clearance of molecular debris and damaged cells lower the organismal inflammatory burden, likely generating antipsychotic effects [209,210]. Conversely, dysfunctional autophagy and accumulation of cellular remains at the gut barrier and BBB may promote inflammation and microbial translocation into the host systemic circulation [66]. This may explain elevated translocation markers and the more diverse blood microbiome documented in SCZ patients [66,70,211].

Gut commensals enjoy immunological tolerance in the GI tract; however, this protection ceases upon migration outside the intestinal barrier, where they can be vehemently attacked by the host immune defenses [212]. Bactericidal antipsychotic drugs likely facilitate the clearance of both translocated microbes and damaged cells (by autophagy activation), lowering the odds of neuroinflammation and psychosis [213,214]. Other non-dopaminergic beneficial effects of antipsychotic drugs, such as telomere elongation, microglial de-escalation, and inhibition of ferroptosis, may likely be explained by the agonist/antagonist AhR binding [6,200,215,216]. For example, AhR antagonists and some partial agonists, including resveratrol, quercetin, or the synthetic compound HBU651, were demonstrated to elongate telomeres, reverse microglial activation, and avert ferroptosis, placing AhR at the epicenter of SCZ pathology [217,218,219,220] (Table 2).

Taken together, the clinical efficacy of antipsychotic drugs may be mediated by both dopaminergic and non-dopaminergic pathways, the latter including lowering neuroinflammation, ferroptosis inhibition, telomere lengthening, and deactivation of microglia [215,221,222,223].

### 5.1. Antipsychotics as Antibacterials

First- and second-generation antipsychotic drugs possess antibacterial properties, suggesting that the elimination of translocated microbes may drive symptomatic relief in psychosis [224]. This is further substantiated by the current efforts to repurpose several antipsychotic drugs as antibiotics [225]. Conversely, antibiotics such as doxycycline and minocycline exert antipsychotic properties, indicating that postsynaptic DA blockade may not be the only mechanism for alleviating psychotic symptoms [226,227].

Other antipsychotic drugs with antimicrobial properties include phenothiazines, compounds capable of eliminating *E. coli,* a bacterium previously associated with SCZ [228,229]. Moreover, haloperidol exerts fungicidal action against *Candida albicans* (*C. albicans*), a BBB-crossing fungus, linked by previous studies to SCZ [230,231,232]. Furthermore, both antipsychotic drugs and IL-22, including the recombinant form, inhibit IFN-γ, a cytokine with established antifungal properties [233,234]. Interestingly, IL-22 exhibits fungicidal action against *C. albicans* as well as antipsychotic-like properties [231,235,236,237,238] (Table 3).

### 5.2. Antipsychotics as Antivirals

Many antipsychotic drugs possess antiviral properties inherited from their parent compound and phenothiazine dye, methylene blue (MB) [251]. For example, chlorpromazine exerts antiviral properties against SARS-CoV-2, the etiologic agent of the COVID-19 pandemic, and is currently in clinical trials (NCT04366739) for this viral infection [252]. Other viruses enter host cells via clathrin-dependent endocytosis (CDE) and are inhibited by several neuroleptics, including chlorpromazine, fluphenazine, perphenazine, prochlorperazine, and thioridazine, emphasizing an alternative, DA-independent mechanisms of action, likely mediated by AhR (functioning as an E3 ubiquitin ligase) [253,254,255].

Several antipsychotic drugs alter the biophysical properties of cell membranes by intercalating themselves into the lipid bilayer, blocking viral fusion with host cells. At the same time, this process may comprise an antipsychotic mechanism by depolarizing neuronal membranes, lowering pathological connectivity and likely calcium entry [256,257]. Moreover, several antipsychotics, including haloperidol, are cationic, amphiphilic compounds that accumulate in lysosomes, likely sabotaging viral replication as well as SCZ-associated lysosomal dysfunction [258,259].

### 5.3. Antipsychotics as Anticancer Drugs

Several first- and second-generation antipsychotic drugs can arrest the cell cycle at the G2/M point, explaining their beneficial effects against some cancers [260,261]. In patients with SCZ, antipsychotic drugs were found to block the paradoxical attempt of mature neurons to re-enter the cell cycle, emphasizing another possible DA-independent mechanism of alleviating psychosis [262,263,264,265,266].

### 5.4. Microbial Phenazines vs. Antipsychotic Phenothiazines

Phenazines are ubiquitous nitrogen-based AhR ligands, released by a wide variety of bacteria, including gut commensals *Pseudomonas* spp. [267,268]. Like phenothiazine antipsychotics, phenazines exhibit anti-inflammatory, anticancer, antimicrobial, and neuroprotective properties, suggesting that they likely bind AhR [269]. Microbial phenazines are natural phenothiazines, generated by the gut commensals to eliminate pathogenic bacteria and malignant cells by ROS production [270,271,272] (Figure 3).

Phenothiazines are MB derivatives which led to the development of the first marketed antipsychotic drug, chlorpromazine, which in 1954 ushered in the era of psychopharmacology [273,274]. Since microbial phenazines are AhR ligands, phenothiazines very likely attach to AhR, suggesting an alternative antipsychotic mechanism [275,276].

## 6. Potential Applications

In this section, we take a closer look at several therapeutic modalities for optimizing intestinal permeability and increasing neurogenesis in both the CNS and the enteric nervous system (ENS). We discuss CAP enhancement, recombinant IL-22, and AhR ligands.

### 6.1. Cholinergic Anti-Inflammatory Pathway Augmentation

In 2005, the Food and Drug Administration (FDA) approved VNS for the treatment of refractory major depressive disorder (MDD). More recent studies have shown that this treatment modality reduces paracellular intestinal permeability, suggesting beneficial effects in both IBD and SCZ [277,278,279]. For example, noninvasive transcutaneous VNS (tVNS), including trans-auricular (taVNS), currently utilized in clinical practice for the treatment of seizure disorder and migraine headaches, has been demonstrated to protect the intestinal barrier by parasympathetic nicotinic action on occludin and zonulin-1 (ZO-1) [280,281,282,283]. In addition, tVNS increases the abundance of beneficial gut microbes, *Bifidobacterium* and *Lactobacillus,* enhancing the gut barrier [284,285,286]. As inflammation negatively affects interoceptive awareness, tVMS may augment insight by increasing CAP function [287,288,289].

A variant of tVNS, ultrasound neuromodulation of the spleen, is based on the recent finding that memory T cells generate ACh in the gut and spleen, influencing microbial diversity [290,291,292]. Moreover, AChE and BuChE inhibitors may accomplish similar results to tVNS by increasing ACh levels [293,294]. A growing body of evidence has connected CAP and α7nAChR with increased neurogenesis in both the adult brain and ENS, emphasizing further central control of the gut barrier [295,296].

Taken together, ACh upregulation via therapeutics or vagal stimulation may have beneficial effects in SCZ by enhancing adult neurogenesis in both the CNS and ENS, lowering microbial translocation.

### 6.2. Recombinant IL-22 for IBD and SCZ

The 2007 conclusion of the Human Microbiome Project and the subsequent finding that the brain and gut AhR/STAT3/IL-22 axis promotes neurogenesis in various niches shedding more light on the gut–brain interaction, suggesting that activating this molecular axis may be beneficial in SCZ [297,298,299,300,301].

IL-22, currently in clinical trials for IBD, exhibits antipsychotic-like properties and promotes thymus gland rejuvenation, suggesting beneficial effects in SCZ (NCT02847052) [302,303,304,305]. To our knowledge, at the time of this writing, there are no studies on IL-22 in SCZ. We have previously hypothesized that F-652, a recombinant human IL-22, can alleviate psychotic symptoms by blocking microbial translocation. Table 2 summarizes the overlapping action mechanisms of IL-22 (F-652) and antipsychotic drugs.

A member of the IL-10 cytokine family, IL-22, interacts with a heterodimeric protein comprising IL-22 and IL-10 receptors, which has been previously implicated in SCZ [305,306,307]. Moreover, IL-22 is an active participant in iron metabolism by inducing transcription of hepcidin and lowering ferroportin, sequestrating intracellular iron in ferritin [308,309]. This is significant as novel studies have shown that iron accumulates in SCZ brains, emphasizing that ferroptosis may be a key player in this pathology [310,311,312]. Recombinant IL-22/F-652 may be beneficial in SCZ as it lowers the risk of ferroptosis by promoting iron storage in ferritin [313].

### 6.3. Dietary and Pharmacological AhR Ligands

Excessive AhR activation was found to disrupt neurogenesis during the development, inducing growth arrest and apoptosis, a pathology encountered in ASD and SCZ [314,315]. Moreover, weight gain, one of the most common adverse effects of clozapine, is likely mediated by AhR and could be averted by this receptor’s antagonist, CH-223191, a likely future metabolic syndrome treatment [316]. However, AhR antagonists also lower IL-22; therefore, studies of partial agonists/antagonists at this receptor are needed as these agents have the ability to selectively target AhR, avoiding adverse effects such as metabolic syndrome or IL-22 downregulation [317,318]. Moreover, selective AhR modulators (SAhRMs), such as flavonoids, may increase IL-22 without excessive AhR activation [318]. For example, quercetin apigenin, luteolin, kaempferol (KPF), and several common spices, including cloves, dill, thyme, nutmeg, and oregano, exert SAhRM properties, indicating selective induction of IL-22 without AhR overactivation [319,320]. Furthermore, synthetic flavonoids alpha-naphtoflavone, 3-methroxy-4-itriflavone, and their derivatives also possess SAhRM properties, suggesting potential benefits in SCZ [321]. AhR was discovered by Alan Poland et al., in 1976 and was characterized as a dioxin receptor [322]. A ligand-activated transcription factor, AhR responds to internal and external stimuli and the research focus has recently shifted from environmental toxins to ligands originating from the intracellular and extracellular body compartments. The discovery that AhR regulates transposable elements (including retrotransposons), DNA segments implicated in SCZ and ASD, has brought this transcription factor closer to the neuropsychiatric arena [323].

AhR has been known to induce cytochrome P450 1A1 (CYP1A1), an enzyme abundantly represented in the kidney, liver, and intestine, involved in phase one metabolism [324]. Approximately 70–80% of all drugs and exogenous molecules are metabolized by CYP1, 2, and 3 enzyme families [325]. CYP1A1 is responsible for the metabolism of fatty acids, steroid hormones, and heavy metals [326]. CYP1A1 was demonstrated to impair the phagocytosis of microorganisms, indicating interference with microbial translocation form the GI tract into the systemic circulation [327].

## 7. Wider Disparate Data and Future Directions

The investigation of how the above links to wider bodies of data should better clarify processes relevant to the pathophysiology and treatment of schizophrenia. As with many neuropsychiatric conditions, suboptimal mitochondrial function is often evident in diverse brain and systemic cells of people with schizophrenia [328]. Notably, AhR and α7nAChR are present on the mitochondrial membrane [329], indicating that direct effects on mitochondria, and therefore on core aspects of cellular function, are likely to be important aspects of pathophysiological alterations in schizophrenia. It is also important to note that AhR, via CYP1A2 and CYP1B1, O-demethylates melatonin to N-acetylserotonin (NAS). Variations in the NAS/melatonin ratio may be important given that NAS is a BDNF mimic via TrkB activation [330] and therefore drives more proliferative responses, whilst melatonin optimizes mitochondrial function and oxidant status, thereby impacting how mitochondrial oxidant production drives ROS-dependent microRNAs, and therefore modulating patterned gene expression. TrkB, like AhR and α7nAChR, is expressed on the mitochondrial membrane [330], indicating that variations in the AhR-driven NAS/melatonin ratio may have significant direct effects on mitochondrial function and thereby on patterned gene expression. It will be important for future research to determine how the mitochondrial presence of AhR, TrkB, and α7nAChR modulate core mitochondrial function and patterned gene expression, and thereby the changes in intercellular fluxes evident in schizophrenia.

This links to wider bodies of data showing increases in pro-inflammatory cytokines and dysregulated HPA axis, leading to increased indoleamine 2,3-dioxygenase (IDO) and tryptophan 2,3-dioxygenase (TDO), driving the raised levels of kynurenine and especially kynurenic acid evident in schizophrenia [331,332]. Both kynurenine and kynurenic acid activate AhR, whilst kynurenic acid may also antagonize the α7nAChR and N-methyl-d-aspartate (NMDA) receptor [332]. Wider immune dysregulation in schizophrenia may therefore be intimately linked to AhR and α7nAChR regulation, with consequences for mitochondrial function and intercellular interactions. Raised levels of systemic pro-inflammatory cytokines increase gut permeability, typically concurrent to gut dysbiosis, indicating how systemic inflammatory processes modulate the gut and the gut’s influence on wider systemic processes, including its interaction with the vagal nerve. Variations in the levels of gut microbiome-derived butyrate may be especially important given the powerful influence of butyrate on mitochondrial function across body cells [333].

## 8. Conclusions

Several SCZ characteristics seem incongruent with DH, while some antipsychotic drugs appear to lower the positive symptoms of SCZ by DA-independent mechanisms. These discrepancies suggest that the third revision of DH may require further adjustment, as the dopaminergic transmission, receiving genetic and epigenetic input, may not be the final step in SCZ pathogenesis. DA is an AhR ligand; therefore, DA-lowering drugs may attenuate psychotic symptoms by indirectly modulating AhR. This seems to indicate that DA may play the role of an upstream effector which alters the function of AhR, the master regulator of exogenous and endogenous input. This may also explain the “delayed onset” of antipsychotic action, despite the fact that DA receptors are blocked very shortly after the dose.

Under physiological circumstances, the gut AhR/STAT3/IL22 axis optimizes microbiota composition and relays the status of “luminal milieu” to IC, directly regulating interoceptive awareness. Recombinant human IL-22/F-652 is an antimicrobial and antiviral molecule that promotes tissue repair, autophagy, and thymic regeneration, reversing SCZ-associated premature cellular and thymic senescence.

Stimulation of VN branches is a noninvasive procedure that promotes beneficial microbiota, lowering local and systemic inflammation. These procedures could correct SCZ-associated anosognosia as well as avert metabolic adverse effects. More AhR studies are needed to clarify this transcription factor’s ligands and their selective actions in a tissue- and context-dependent manner. 

## Figures and Tables

**Figure 1 reports-06-00036-f001:**
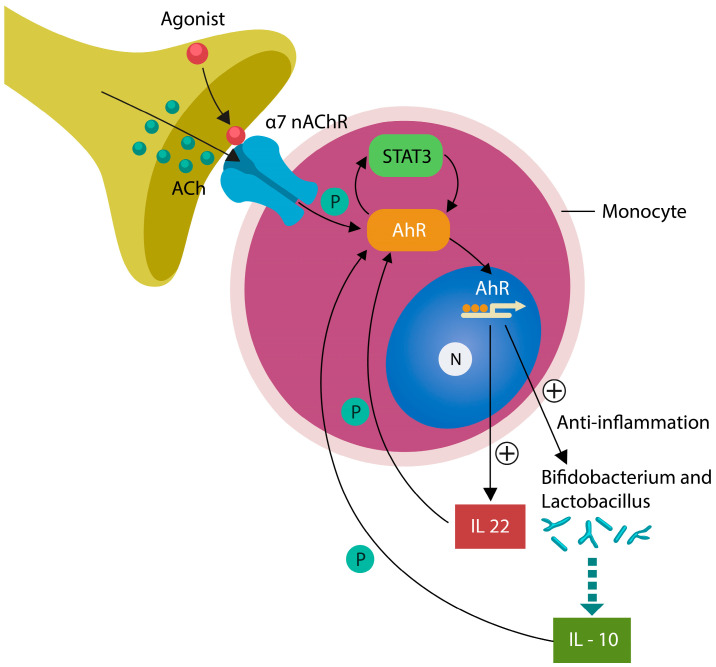
Upregulated peripheral monocytes may reflect the status of IC-mediated interoceptive awareness. At the molecular level, insight is likely driven by α7nAChRs which connect CAP to the AhR/STAT3/IL-22 axis. Beneficial gut microbes *Bifidobacterium* and *Lactobacillus* release IL-10, a cytokine that provides feedback to the IC via STAT3 phosphorylation. IL-22 (related to IL-10) also phosphorylates STAT3, closing the feedback loop.

**Figure 2 reports-06-00036-f002:**
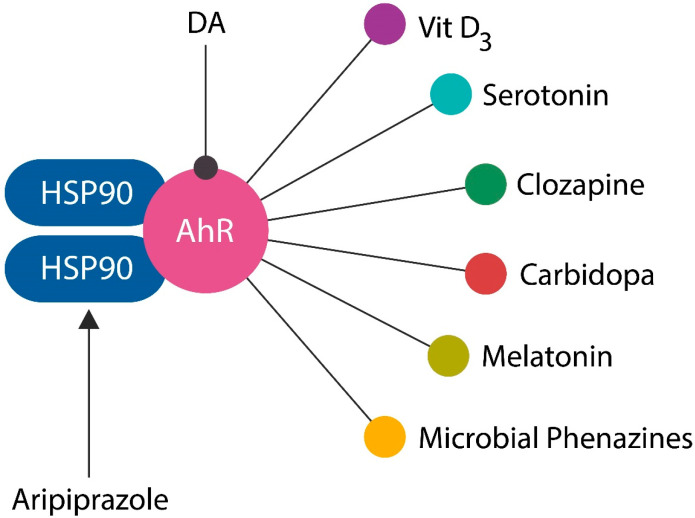
AhR ligands associated with SCZ. DA is an AhR ligand, while aripiprazole binds HSP90. AhR agonists, vitamin D3, melatonin, and serotonin likely account for the increasing prevalence of SCZ with latitude, while pollutants, including dioxin, correlate with the increased prevalence of SCZ in urban environments. Microbial phenazines are natural phenothiazines, suggesting that these drugs are likely AhR. ligands.

**Figure 3 reports-06-00036-f003:**
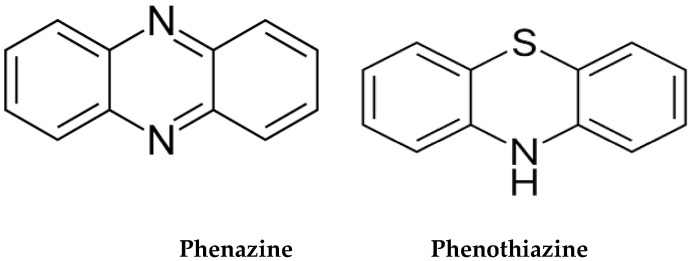
Structural similarity between microbial phenazines and phenothiazines. In the gut, *Pseudomonas aeruginosa* is the main producer of phenazines. Phenazines upregulate ACh by inhibiting its degrading enzymes, acetylcholinesterase (AChE) and butyryl acetylcholinesterase (BChE), enhancing cholinergic transmission.

**Table 1 reports-06-00036-t001:** DA-unrelated SCZ characteristics explained by translocated microbes/AhR activation.

DH-Discordant SCZ Features	Non-DA Mechanisms	References
Negative symptoms	Translocation of *Hafnei alvei*, *Pseudomonas aeruginosa*, *Morganella morganii*, *Pseudomonas putida*, and *Klebsiella pneumoniae*	[12,13,14]
Comorbidity with IBD	AhR/STAT3/IL-22-regulated intestinal permeability and microbiota translocation	[11,15,16]
Comorbidity with HIV	AhR/STAT3/IL22-regulateted gut barrier permeability	[12,17,18]
Poor insight (anosognosia)	IC activation by gut *Prevotella* and *Bacteroides* abundance	[19,20,21,22,23,24,25]
Higher prevalence in urban areas	Pollutants are AhR ligands associated with SCZ and are more prevalent in industrialized countries and urban areas	[26,27,28,29,30,31,32,33,34]
Increasing prevalence with the distance from the equator	Sunlight-driven vitamin D derivatives and tryptophan light metabolites are AhR ligands	[35,36,37,38,39]
Autoantibodies	Gut microbes express molecules, including GABA and NMDA, which can elicit formation of antibodies upon translocation	[40,41,42]

**Table 2 reports-06-00036-t002:** Exogenous and endogenous AhR agonists and antagonists.

Exogenous Agonists	Endogenous Agonists	Dietary Antagonists	Pharmacological Antagonists
Benzotriazole UV stabilizer	Tryptophan photo metabolites and FICZ	quercetin	CH-223191
Plasticizers (Bisphenol)	Indoles	apigenin	Alpha-naphtoflavone 3-methroxy-4-itriflavone
Clozapine	D3 hydroxyderivatives	luteolin	BAY2416964
Carbidopa	DA	kaempferol	HBU651

**Table 3 reports-06-00036-t003:** Antipsychotic-like properties of IL22 (or recombinant IL22).

Antipsychotic Drugs	Recombinant IL-22	References
JAK-STAT activation	JAK-STAT activation	[239,240]
Neuroprotective	Neuroprotective	[241,242]
IFN-γ inhibitor	IFN-γ inhibitor	[243,244]
Activate autophagy	Activate autophagy	[245,246,247]
Antibacterial/antiviral	Antibacterial/antiviral	[230,248,249,250]

## Data Availability

Not applicable.

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
