# Peer review of "Six Decades of Dopamine Hypothesis: Is Aryl Hydrocarbon Receptor the New D2?"

_reports, 2023, doi:10.3390/reports6030036_

Round 1

Reviewer 1 Report

This is a comprehensive review of how the classical dopamine hypothesis of schizophrenia may be adapted to include interactions with the aryl hydrocarbon receptor (AhR), and the implications that this would have for the conceptualization and treatment of people classed with schizophrenia. The authors highlight an important role of the gut-brain axis and the role of the vagal inputs to intestinal epithelial cells, emphasizing the importance of the AhR/STAT3/IL22 axis.

The authors make a reasonable attempt to integrate variations of the 1960’s ‘dopamine hypothesis’ of schizophrenia with more recent data on cellular and systemic systems. However, this clearly suffers from selective abstraction, which could be rectified by adding a section before the ‘’Conclusions” Section e.g.

“Wider Disparate Data

The investigation of how the above links to wider bodies of data should better clarify processes relevant to the pathophysiology and treatment of schizophrenia. As with many neuropsychiatric conditions, suboptimal mitochondrial function is often evident in diverse brain and systemic cells of people with schizophrenia [Fizíková et al., 2023]. Notably, the AhR and α7nAChR are present on the mitochondrial membrane [Anderson and Maes, 2017], indicating that direct effects on mitochondria, and therefore on core aspects of cellular function, are likely to be important aspects of pathophysiological alterations in schizophrenia. It is also important to note that the AhR, via CYP1A2 and CYP1B1, O-demethylates melatonin to N-acetylserotonin (NAS) [Mokkawes and de Visser, 2023]. Variations in the NAS/melatonin ratio may be important given that NAS is a BDNF mimic, via TrkB activation [Jang et al., 2010] and therefore drives more proliferative responses, whilst melatonin optimizes mitochondrial function and oxidant status, thereby impacting on how mitochondrial oxidant production drives ROS-dependent microRNAs, and therefore modulating patterned gene expression. TrkB, like the AhR and α7nAChR, is expressed on the mitochondrial membrane [Anderson and Maes, 2017], indicating that variations in AhR-driven NAS/melatonin ratio may have significant direct effects on mitochondrial function and thereby on patterned gene expression. It will be important for future research to determine how the mitochondrial presence of the AhR, TrkB, and the α7nAChR, modulate core mitochondrial function and patterned gene expression, and thereby the changes in intercellular fluxes evident in schizophrenia.

This links to wider bodies of data showing increases in pro-inflammatory cytokines and dysregulated HPA axis, leading to increased indoleamine 2,3-dioxygenase (IDO) and tryptophan 2,3-dioxygenase (TDO), driving the raised levels of kynurenine and especially kynurenic acid  evident in schizophrenia [Hare et al., 2023]. Both kynurenine and kynurenic acid activate the AhR, whilst kynurenic acid may also antagonize the α7nAChR and N-methyl-d-aspartate (NMDA) receptor [Hare et al., 2023]. Wider immune dysregulation in schizophrenia may therefore be intimately linked to AhR and α7nAChR regulation, with consequences for mitochondrial function and intercellular interactions. Raised levels of systemic pro-inflammatory cytokines increase gut permeability, typically concurrent to gut dysbiosis, indicating how systemic inflammatory processes modulate the gut and the gut’s influence on wider systemic processes, including its interaction with the vagal nerve. Variations in the levels of gut microbiome-derived butyrate may be especially important given the powerful influence of butyrate on the mitochondrial function across body cells [Anderson and Maes, 2020]. “

The inclusion of something like the above would give the reader links to wider bodies of data and may encourage more imaginative conceptualizations and treatments for this poorly conceptualized and managed condition.

References

Fizíková I, Dragašek J, Račay P. Mitochondrial Dysfunction, Altered Mitochondrial Oxygen, and Energy Metabolism Associated with the Pathogenesis of Schizophrenia. Int J Mol Sci. 2023 Apr 28;24(9):7991. doi: 10.3390/ijms24097991. PMID: 37175697; PMCID: PMC10178941.

Anderson G, Maes M. Interactions of Tryptophan and Its Catabolites With Melatonin and the Alpha 7 Nicotinic Receptor in Central Nervous System and Psychiatric Disorders: Role of the Aryl Hydrocarbon Receptor and Direct Mitochondria Regulation. Int J Tryptophan Res. 2017 Feb 16;10:1178646917691738. doi: 10.1177/1178646917691738. PMID: 28469467; PMCID: PMC5398327.

Mokkawes T, de Visser SP. Melatonin Activation by Cytochrome P450 Isozymes: How Does CYP1A2 Compare to CYP1A1? Int J Mol Sci. 2023 Feb 11;24(4):3651. doi: 10.3390/ijms24043651. PMID: 36835057; PMCID: PMC9959256.

Jang SW, Liu X, Pradoldej S, Tosini G, Chang Q, Iuvone PM, Ye K. N-acetylserotonin activates TrkB receptor in a circadian rhythm. Proc Natl Acad Sci U S A. 2010 Feb 23;107(8):3876-81. doi: 10.1073/pnas.0912531107. Epub 2010 Feb 4. PMID: 20133677; PMCID: PMC2840510.

Hare SM, Adhikari BM, Mo C, Chen S, Wijtenburg SA, Seneviratne C, Kane-Gerard S, Sathyasaikumar KV, Notarangelo FM, Schwarcz R, Kelly DL, Rowland LM, Buchanan RW. Tryptophan challenge in individuals with schizophrenia and healthy controls: acute effects on circulating kynurenine and kynurenic acid, cognition and cerebral blood flow. Neuropsychopharmacology. 2023 Apr 28. doi: 10.1038/s41386-023-01587-3. Epub ahead of print. PMID: 37118058.

Anderson G, Maes M. Gut Dysbiosis Dysregulates Central and Systemic Homeostasis via Suboptimal Mitochondrial Function: Assessment, Treatment and Classification Implications. Curr Top Med Chem. 2020;20(7):524-539. doi: 10.2174/1568026620666200131094445. PMID: 32003689.

Author Response

File attached

Reviewer 2 Report

The authors tried to develop new hypothesis about the pathophysiology of schizophrenia beyond dopamine theory. Because they focused on the AhR/STAT3/IL-22 axis, this manuscript promotes our understanding of schizophrenia. However, there are several issues to be clarified.

Inappropriate citation are found in this manuscript.

1)     The authors cited the paper by Borovikova et al., in the beginning of “cholinergic anti-inflammatory pathway and insight” (page 6). However, there is no citation of Borovikova’s paper in References. Apparently, this citation is important to develop AhR hypothesis. Please show the citation of this paper in References, and confirm whether alpha7 nAChR is involved in the activation of AhR.

2)     The authors only cited Ref. 11 to demonstrate the comorbidity of schizophrenia with IBD. This phenomenon is not globally approved in the psychiatric field. Why they did not cite the following papers in which the prevalence of IBD in patients with schizophrenia?

Sung K-Y et al., Aliment Pharmacol Ther 55:1192-1201, 2022.

Bernstein CN, et al., Inflamm Bowel Dis 25:360-368, 2019.

Ann Marie R, et al., J Psychosom Res 101:17-23, 2017.

3)     In Table 1, Ref. 158-160, 162, and 163 are shown in the connection of “Comorbidity of IBD”. But, these papers did not mention the pathophysiology of schizophrenia. How do these papers support the evidence of the comorbidity?

4)     In the section “AhR, and antipsychotic drugs” (page 9), the authors demonstrated that the transcription factor, AhR, regulated the expression of the SCZ risk genes, and cited Ref. 175-177. However, these papers did not mention the SCZ risk genes.

5)     Why did the authors avoid to mention the following paper in which the direct involvement of AhR in the pathophysiology of schizophrenia?

Schubert K O et al., Schizophr Res 167:64-72, 2015.

Please revise Visual abstract or Figure 1. In Visual abstract, the signal from alpha7 nAChR activates AhR and AhR binds its specific binding site at the promoter of the IL-22 gene. However, in Figure 2, the signal from alpha7 nAChR activates STAT3 and STAT3 binds its specific binding site at the promoter of the IL-22 gene. There is a discrepancy between Visual abstract and Figure 1. 

Author Response

n

Reviewer 3 Report

The article presented for assessment has some shortcomings which should be corrected before publication

1. There is no author affiliation on the title page

2. There should be a graphic abstract, not Visual

3. The correct abbreviation for interleukin 22 is IL-22. please correct throughout the manuscript. the same applies to IL-10

4. Line 102. citation 554 in references is missing

5. Table 1. please correct the spelling of receptors to the correct AhR.

6. Please write a chapter on the structure and function of the AhR receptor.

7. Please write a chapter on the regulation of cytochrome P450 enzymes by the AhR receptor following administration of neuroleptics. 

8. There are numerous typos and errors in the text, please proofread the language. 

There are numerous typos and errors in the text, please proofread the language. 

Round 2

Reviewer 2 Report

It seems that the Ach/AhR/STAT#/IL-22 axis playa an important role in this aryl hydrocarbon receptor hypothesis of schizophrenia. The authors cited Refs. 131-133 to establish the molecular mechanism of this hypothesis. However, there is no evidence showing this axis. For example, Figure 3 in Ref. 132 showed that AhR induced IL-22 expression in cooperation with RoR gamma, but not STAT3, in Th 17 cells. AhR reported to inhibit IL-2 expression by inducing the epigenetic modifier Aiolos in cooperation with STAT3 in Th 17 cells. In contrast, Chen Y and colleagues (Cell Death Discovery 8:141, 2022) demonstrated that alpha-7 nicotinic acetylcholine receptor agonist inhibited STAT3 signal pathway. In this context, the molecular mechanism of this hypothesis is uncertain. Please show the appropriate evidences of this hypothesis and revise this manuscript.    

no problem 

Author Response

Please see attached,

Reviewer 3 Report

Now this manuscript is ready for further process of publication.

Round 3

Reviewer 2 Report

The author appropriately responded to comments. However, the references in this revised manuscript seems to be inappropriate. Please cite the papers in this response letter to comments in suitable sections. 
